# ELENAS: Elementary Neural Architecture Search

Lukas Aichberger[1]  Günter Klambauer[1]

[1] ELLIS Unit Linz and LIT AI Lab, Institute for Machine Learning, Johannes Kepler University Linz, Austria

**Abstract**  Deep neural networks typically rely on a few key building blocks such as feed-forward, convolution, recurrent, long short-term memory, or attention blocks. On an elementary level, these blocks consist of a relatively small number of different mathematical operations. However, as the number of all combinations of these operations is immense, crafting such novel building blocks requires profound expert knowledge and is far from being fully explored. We propose Elementary Neural Architecture Search (ELENAS), a method that learns to combine elementary mathematical operations to form new building blocks for deep neural networks. These building blocks are represented as computational graphs, which are processed by graph neural networks as part of a reinforcement learning system. Our approach contrasts the current research direction of Neural Architecture Search, which mainly focuses on designing neural networks by altering and combining a few, already established, building blocks. In a set of experiments, we demonstrate that our method leads to efficient building blocks that achieve strong generalization and transfer well to real-world data. When stacked together, they approach and even outperform state-of-the-art neural networks at several prediction tasks. Our underlying methodological framework offers high flexibility and broad applicability across domains while requiring relatively small computational costs. Consequently, it has the potential to find novel building blocks that become of general importance for machine learning practitioners beyond specific use cases.

## 1 Introduction

Over the past years, novel neural architectural building blocks have been developed and used in artificial neural networks, which vastly accelerated the progress of deep learning (Paszke et al., 2019; Krizhevsky et al., 2017). Neural Architecture Search (NAS) has successfully combined these building blocks – e.g. various activation functions, normalization layers, pooling layers, and convolution layers with different filter sizes – to form complex, deep neural networks that outperform existing human-designed neural networks on specific datasets, notably ImageNet (Zoph et al., 2018; Liu et al., 2018; Real et al., 2018; Hu et al., 2020). Recently, the overall research progress in NAS is growing considerably (Elsken et al., 2019; Ren et al., 2020; White et al., 2023).

The building blocks used in the majority of NAS approaches have mainly been derived by hand and are generally challenging to design. For instance, experts formulated assumptions about how cell states are able to incorporate a long short-term memory (LSTM) (Hochreiter and Schmidhuber, 1997) or how attention mechanisms have the ability to narrow down relevant features (Vaswani et al., 2017). It can be observed that these key building blocks usually combine a relatively small number of underlying elementary mathematical operations such as basic arithmetic operations for scalar and matrix inputs, as well as non-linear activation functions (Deisenroth et al., 2020). These operations are combined in a specific way to form such building blocks (Goodfellow et al., 2016). This implies that the combinations of these operations form a highly multidimensional space that is far from being fully explored.

In contrast to well-established search spaces in NAS, the majority of building blocks included in this search space of elementary operations are not valid and thus cannot be used in neural

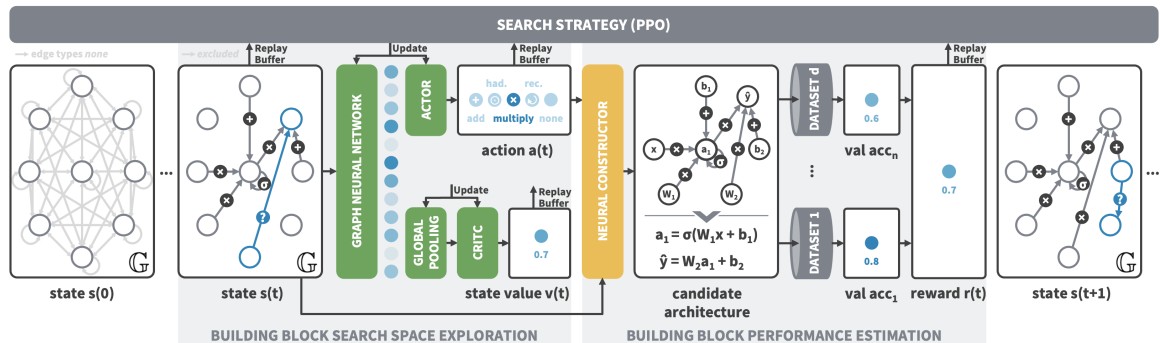

Figure 1: **Overview**. ELENAS utilizes the search strategy PPO to design neural architectural building blocks represented by computation graphs $\mathbb{G}$ (state). It is sequentially constructed by predicting the type of all edges (action), thereby exploring the search space of elementary mathematical operations. The general performance of the building block (reward) is estimated on synthetic datasets of increasing complexity.

networks. Not only is an exhaustive manual search in this vast combinatorial space infeasible but it also requires specific search strategies to explore and reason from this very general search space. However, our hypothesis is that within this underexplored search space, there are several undiscovered novel building blocks, akin to LSTM or attention blocks, that can be integrated into deep neural networks to improve their performance in an entire domain.

To address these challenges when searching for novel building blocks by combining elementary mathematical operations, we propose Elementary Neural Architecture Search (ELENAS). Our method, illustrated in Figure 1, leverages reinforcement learning (RL) (Sutton and Barto, 2018) and specifically adopts the firmly established proximal policy optimization (PPO) algorithm (Schulman et al., 2017) to design unprecedented building blocks represented as computational graphs. Thereby, the RL agent employs a graph neural network (GNN) (Bronstein et al., 2021) to synergistically incorporate expertise in both combining these elementary mathematical operations as well as designing the topology of computation graphs. Our method proposes an encoding of the computation graph that addresses the complexity of the search space and improves the search process. The resultant building blocks are evaluated on synthetic datasets of increasing levels of difficulty that assess their general capabilities in the respective domain. By circumventing the necessity to apply them on complex real-world data, this approach requires only nominal computational resources. This leads to previously unparalleled building blocks that generalize well and – when stacked together to tailor the neural network for a particular task – approach or even outperform existing modern neural networks in the specified domain while requiring only a fraction of the learnable parameters.

Our main contributions can be summarized as follows:

(i) We propose a general search space of elementary mathematical operations that allows for high flexibility in both the combinations of operations and the topology of neural architectural building blocks. This search space includes a variety of unprecedented building blocks, as it induces as little as possible domain expertise.

(ii) We introduce ELENAS for designing general-purpose building blocks by utilizing this search space of elementary mathematical operations and evaluating them on specific synthetic tasks. Our method trains a reinforcement learning agent consisting of graph neural networks to search for and learn from computation graphs of the respective building blocks.

(iii) We present the building blocks ELENA1 and ELENA2 for the domain of sequential data. They have been designed from elementary mathematical operations by our method, are parameter-efficient, achieve strong generalization, and transfer well to real-world data.

## 2 Related Work

Current research in NAS mainly focuses on the field of computer vision (Elsken et al., 2019; Ren et al., 2020; White et al., 2023), with successes primarily in finding high-performing convolutional neural networks (CNNs). Recent work usually combines already established building blocks to form higher-order topological structures that incrementally increase the performance on a given task (Baker et al., 2016; Liu et al., 2017; Zoph et al., 2018; Cai et al., 2018; Xu et al., 2019; Ru et al., 2020; Roberts et al., 2021). These building blocks are commonly defined as triplets of fixed normalization, fixed activation, and convolution operations of varying filter size (White et al., 2023).

Few of these approaches additionally searched for recurrent neural networks (RNNs) (Klyuchnikov et al., 2022). Pham et al. (2018), in particular, computes the first node representation $\mathbf{h}_1$ of the recurrent cell by sampling an activation function $f(\cdot)$ and then combining the input $\mathbf{x}(t)$ at the current time step with the cell output representation $\mathbf{h}(t-1)$ from the previous time step, both transformed by learnable parameters $\mathbf{W}$: $\mathbf{h}_1(t) = f(\mathbf{W}^x\,\mathbf{x}(t) + \mathbf{W}_1^h\,\mathbf{h}(t-1))$. Subsequently, for a predefined number of nodes in the cell, a node representation $\mathbf{h}_i(t)$ from the current time step is sampled together with an activation function $f(\cdot)$ to compute the representation: $\mathbf{h}_j(t) = f(\mathbf{W}_j^h\,\mathbf{h}_i(t))$. This method is similar to the one from Zoph and Le (2016) and Liu et al. (2018), which fix the topology of the recurrent cell but instead sample an operation together with an activation function to compute a node. However, these methods usually incorporate severe expert knowledge to construct neural architectures, thus limiting and biasing the possible outcome. Inspired by the LSTM architecture (Hochreiter and Schmidhuber, 1997), Zoph and Le (2016) introduce cell variables that already represent memory states, and Pham et al. (2018) as well as Liu et al. (2018) add fixed highway connections when computing a node representation: $h_j(t) = \mathbf{c}_j(t) \odot f(\mathbf{W}_j^h\,\mathbf{h}_i(t)) + (1 - \mathbf{c}_j(t)) \odot \mathbf{h}_i(t)$, with $\mathbf{c}_j(t) = \mathrm{sigmoid}(\mathbf{W}_j^c\,\mathbf{h}_i(t))$ and $\odot$ denoting the Hadamard product. The best neural architectures that resulted from these approaches are compared to the ones designed by ELENAS in Section 4.

Nonetheless, there is scarce literature on utilizing elementary mathematical operations to search for general-purpose neural architectural building blocks. The work most closely to our method has been proposed by Schrimpf et al. (2017), searching for RNNs by generating a tree of elementary operations. However, they considerably constrain the set of operations and their possible combinations. Similarly to the work by Zoph and Le (2016), the nodes that are combined by these operations incorporate severe domain expertise, as they already include long-term memory by design (Schrimpf et al., 2017). A few other approaches exist that are closely related from the perspective of utilizing elementary operations, but that search for different components. For instance, Liu et al. (2020) search for normalization layers and activation functions by leveraging element-wise operations such as addition, multiplication, the sigmoid function, and aggregation operations. Their approach led to normalization layers with unique structures that generalize well across various tasks (Liu et al., 2020). Similarly, Ramachandran et al. (2017) used primitive operations to search for activation functions. Their work resulted in the well-known Swish activation function (Ramachandran et al., 2017). Finally, Real et al. (2020) search for comprehensive machine learning algorithms, with modern techniques emerging from basic mathematical operations.

Although our approach contrasts the current research direction of NAS, it builds upon some of the promising concepts of recent advances in the field. For instance, Zoph et al. (2018) propose the popular cell-based search space to search for cells that are stacked in a predefined manner to form the overall, chain-structured, neural network, which has been widely adopted (Ying et al., 2019; Liu et al., 2018). Their work also utilizes PPO (Schulman et al., 2017), which they found to be faster and more stable than the REINFORCE policy gradient algorithm (Zoph and Le, 2016). Moreover, the performance of the candidate architectures is evaluated on a smaller proxy dataset, similar to our performance estimation strategy. In this way, their highest-performing cells outperform the best human-invented neural networks on unseen datasets (Zoph et al., 2018).

## 3 ELENAS

At the core of our method is the search for general-purpose neural architectural building blocks using elementary mathematical operations that can be utilized as part of deep neural networks (see overview in Figure 1). Such a building block can be represented as a computation graph $\mathbb{G}$ (see Section 3.1). In general, the process of designing a building block can be reduced to exploring the search space of elementary mathematical operations (Section 3.2), namely predicting each of the edge types in $\mathbb{G}$. We utilize the RL algorithm PPO (Schulman et al., 2017) as the search strategy (see Section 3.3). $\mathbb{G}$, namely the state $s(t)$ in the RL setting, is embedded by a GNN and a graph global pooling. The type of a given edge in $\mathbb{G}$, namely the action $a(t)$ in the RL setting, is then predicted based on this embedding. This results in a new state that is converted to a neural architectural building block by the neural constructor. The general performance estimation of the building block (see Section 3.4) is conducted on synthetic datasets of increasing complexity in the respective domain. The validation accuracies together with the topology of $\mathbb{G}$ contribute to the reward $r(t)$ in the RL setting. This process is repeated for every edge in $\mathbb{G}$, composing one episode on which the PPO agent is trained on to incorporate expertise in designing high-performing building blocks.

### 3.1 Building Block Representation

**Nodes and edges in** $\mathbb{G}$. As mentioned above, a building block is represented as a computation graph $\mathbb{G}$ (Hamilton, 2020), in which nodes symbolize learnable parameters or (latent) representations, and directed edges in $\mathbb{G}$ symbolize elementary mathematical operations (Liu et al., 2018). Concretely, we use four different node types: *input*, *initial*, *intermediate*, and *final* nodes. Node $n_0$ is defined as the input node and node $n_i$ is defined as initial if $i$ is uneven. Input and initial nodes can only have outgoing but no incoming connections or self-connections, while all other nodes can have incoming, outgoing, and self-connections. We define ten different edge types as part of the search space below. Additionally to the four main node types and the then edge types, we define specific information resulting from the topology of $\mathbb{G}$. Regarding nodes, these are *recursion*, *unused*, and *dead*. A *recursion* node stores a (latent) representation of a node from the previous time step. A *unused* or *dead* node is not connected directly or indirectly to $n_0$ with edges of type other than *none*. Regarding edges, these are *dead* and *backwards*. The former refers to edges of type other than *none* that do not connect $n_0$. The latter refers to artificially added incoming edges to *input* or *initial* nodes, thus are not to be predicted but only serve to improve the information flow in $\mathbb{G}$ when being processed. Also, we empirically observed that adding the information *position* of a node in $\mathbb{G}$ – with the use of trigonometric functions – as well as of an edge in $\mathbb{G}$ – namely whether an edge has already been predicted – substantially improves the overall search process.

**Size of** $\mathbb{G}$. We restrict the number of total nodes $m$ in $\mathbb{G}$ to 32. However, our method also allows for increasing $m$ to dynamically adapt to the search progress. For $m$ nodes, the total number of edges is equal to $\frac{m(m-1)}{2}$ in case $m$ is uneven and $\frac{m(m-2)}{2}$ in case $m$ is even, since each intermediate node has an incoming connection from all nodes in $\mathbb{G}$, including self-connection. This implies that the length of an episode of designing a building block is equal to the number of edges in $\mathbb{G}$, with each edge having the same number of possible actions and the number of edges increasing quadratically with the number of nodes. Thereby, the order in which edges are predicted can be chosen arbitrarily. We experimentally determined that predicting edges according to sub-graphs in $\mathbb{G}$ improves the search for building blocks, addressed in more detail in Section F of the appendix.

### 3.2 Search space

The search space contains elementary mathematical operations assigned to the edge types within the computation graph $\mathbb{G}$. Our approach reduces the search space by a factor of two without compromising its generality. Depending on the nodes connected by an edge, we define different permissible operations, named *node combining operations* and *node transforming operations*.

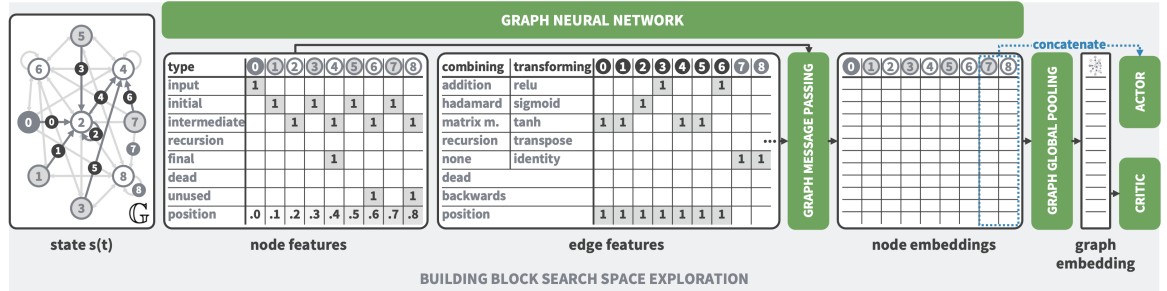

Figure 2: **Building block search space exploration**. ELENAS utilizes a GNN to process the computation graph $\mathbb{G}$. Thereby, node features encode the predefined node types and edge features encode the elementary mathematical operation that the edges represent. The actor computes edge type probabilities based on the concatenated node embeddings of the two nodes that the respective edge connects. The critic computes the state value based on a single graph embedding that is computed by globally pooling all node embeddings.

**Node combining operations**. They refer to a type of edge that connects a node with a different node. These operations take two or more node representations to compute the representation of an additional node. We chose the five frequently used operations *addition, matrix multiplication, Hadamard product* (also referred to as element-wise multiplication), *recursion*, and *none* (implying that no connection is present between the two respective nodes). In case recursive connections are present in $\mathbb{G}$, the input is processed sequentially and the recursion operator feeds the (latent) representation of a node at a time step into a node at the subsequent time step (Goodfellow et al., 2016; Hamilton, 2020). A recursion operator as such has not yet been present in the current research of NAS, making it a valuable contribution of our proposed method.

**Node transforming operations**. They refer to a type of edge that connects a node with itself (equivalently called self-connection). These operations take a single node representation and apply an in-place transformation. We again chose the five frequently used operations *sigmoid* activation function, *tanh* activation function, *relu* activation function, *transpose*, and *identity*.

The generality of this approach is preserved as node combining operations are impractical when applied to a single node representation, and the same is valid for node transforming operations applied on more than one node representation. It has to be noted that the number of transforming operations must be equal to the number of combining operations. Apart from that, our methodological framework is not limited to the chosen operations. Since the elementary mathematical operations serve a general purpose, the expressiveness of this cell-based search space can be preserved more decisively compared to previous work (Zoph et al., 2018; Ying et al., 2019; Liu et al., 2018). For instance, both the LSTM block (Hochreiter and Schmidhuber, 1997) and the attention block (Vaswani et al., 2017) are part of this search space.

### 3.3 Search strategy

**Graph Neural Network (GNN)**. As part of the RL-based PPO search strategy, the computation graph $\mathbb{G}$, representing the state, is processed by a GNN. Given node features that encode the node types and edge features that encode the elementary mathematical operations, the GNN outputs a node embedding for each node in $\mathbb{G}$, as illustrated in Figure 2. The GNN consists of message-passing layers proposed by Mo et al. (2021), an extension of the Graph Attention Network (GAT) by Veličković et al. (2017). These layers achieved the highest performance on an artificially created self-supervised task among six message-passing layers (Veličković et al., 2017; Shi et al., 2020b; Brody et al., 2021; Rozemberczki et al., 2020; Li et al., 2020), detailed in Section F of the appendix.

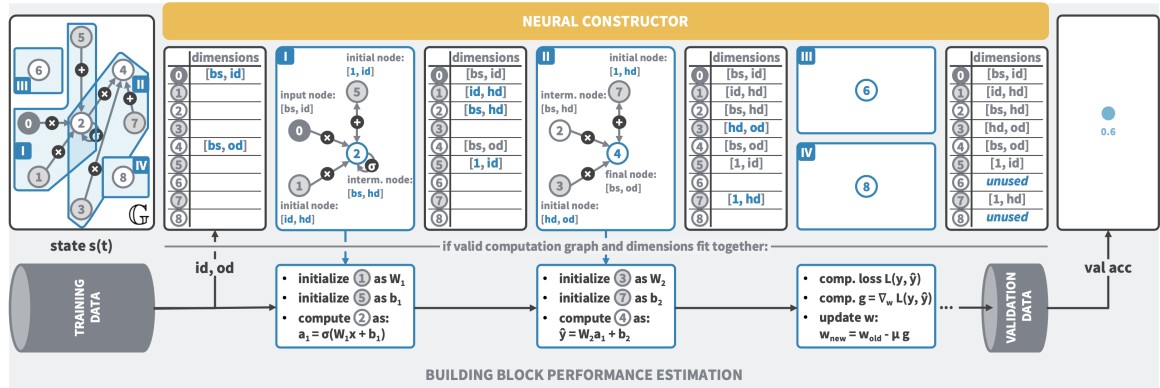

Figure 3: **Building block performance estimation**. ELENAS utilizes the 'neural constructor' to transform the computation graph $\mathbb{G}$ to a neural architectural building block. Therfore, the dimensions of the (latent) representation of each intermediate node and all nodes used to compute it, are determined. Also, the actual computation is defined based on the topology and edges of $\mathbb{G}$. If $\mathbb{G}$ is valid and the dimensions fit together, all initial nodes are randomly initialized and the building block is trained and evaluated on the provided dataset.

**Actor-Critic Network**. The RL algorithm PPO is based on the actor-critic architecture that builds upon the output of the GNN. The actor, a feed-forward neural network, predicts the edge-type probabilities based on the concatenated node embeddings of the two nodes that the respective edge connects. The edge types represent the possible actions of the RL setting. The critic first applies a graph global pooling on all node embeddings to receive a single graph embedding. This graph embedding is to incorporate the relevant information of the graph to sample the state value $v(t)$ from (Baek et al., 2021; Vinyals et al., 2015). Thus, we chose the graph global pooling method *Graph Multiset Transformer* by Baek et al. (2021). Compared to other methods, this multi-head attention-based global pooling satisfies both injectiveness and permutation invariance and, therefore, theoretically is as powerful as the Weisfeiler-Lehman graph isomorphism test Weisfeiler and Leman (1968). A feed-forward neural network of the critic then predicts the state value, an estimate of the goodness of the given state (Sutton and Barto, 2018), based on the graph embedding.

**Search process**. Starting with the initial state that consists of edges of type *none* or *identity* only, the state value is computed together with an edge type for each edge in $\mathbb{G}$, the state is updated, and the reward is computed as described below. Old state, new state, state value, edge, action, action probabilities, and reward are stored in a reply buffer. Once the replay buffer reaches a predefined size, the RL agent is updated. The loss components are computed according to the PPO algorithm and the RL agent is optimized using mini-batch stochastic gradient descent (Sutton and Barto, 2018) for a predefined number of epochs. Thereby, it learns to prefer actions that lead to a high cumulative reward, thus to high-performing building blocks. The reply buffer is then emptied and the search process is continued.

**Alternative search strategies**. While ELENAS employs RL for its search strategy, it is not the only viable option. Other black-box optimization techniques such as Bayesian optimization could also be applied, potentially using a graph neural network as the surrogate model (Ru et al., 2020; Shi et al., 2020a). However, it seems unavoidable to comprehend the properties of the elementary mathematical operations, as the big search space only includes a relatively low fraction of high-performing building blocks. Hence, employing a RL agent that acquires knowledge from graph structures and consequently refines the search space, appears to be a natural choice. As a result, search strategies such as random or local search (Chen et al., 2018; Siems et al., 2020), and evolutionary or genetic algorithms (Real et al., 2018), have not been the fist choice for addressing the search space at hand.

We opted to employ the PPO algorithm for our search strategy due to its robust theoretical underpinnings, as well as its superior speed and stability compared to other policy gradient algorithms. (Schulman et al., 2017; Zoph and Le, 2016; Zoph et al., 2018).

### 3.4 Performance estimation

**Neural constructor**. After applying an action to the state and thus potentially modifying the computation graph $\mathbb{G}$, it has to be transformed into a neural architecture to estimate the performance of the underlying building block, as illustrated in Figure 3 and detailed in Section D of the appendix. This evaluation is carried out sequentially on pre-defined synthetic datasets $d \in \mathcal{D}$ of increasing complexity, while training is terminated if the validation accuracy $a_d^{val}$ fails to meet a dataset-specific threshold. By this means, only building blocks that demonstrate capability in learning simple tasks are evaluated in progressively more difficult tasks to conserve computational resources.

**Reward function**. The total reward $r(t)$ is based on $a_d^{val}$ that the building block achieved on the specific dataset $d \in \mathcal{D}$, with $a_d^{avg}$ indicating the average accuracy by random initialization, $s_d$ representing the factor by which the reward for this dataset $d$ is weighted, and $t$ the specific step in an episode of designing the building block:

$$r(t) = \sum_{d \in \mathcal{D}} \text{ReLU}\Big(\exp\big(a_d^{val}(t)\big) - \exp\big(a_d^{avg}(t)\big)\Big) * s_d + r_{shape}(t) \tag{1}$$

Additionally, $r(t)$ is based on reward shaping $r_{shape}(t)$. We empirically observed that this considerably influences the search process, as the RL agent initially has to learn to avoid predicting edge types that result in invalid or inefficient $\mathbb{G}$. Therefore, whenever a building block could successfully be trained on the first task but achieved a validation accuracy below the threshold, an additional positive reward is assigned. This cumulatively expresses that $\mathbb{G}$ is valid (i.e. no illegitimate recursions in the graph, the input node is used in the computation, and there is a proper final node) and that the dimensions fit together. Moreover, for more than two incoming edges of type *matmul*, *hadamard*, and *add*, as well as for more than one incoming edge of type *recursion*, an additional negative reward is assigned. This specific reward shaping allows for a more stable search process and is addressed in more detail in Section E of the appendix.

## 4 Experiments

We demonstrate that ELENAS is capable of designing novel building blocks explicitly in the domain of sequential data, with the above-defined search space, search strategy and performance estimation strategy. Nonetheless, the methodological framework extends beyond this domain and can be applied to any set of elementary mathematical operations and datasets, making it an adaptable tool for a wide range of applications.

### 4.1 Datasets

For the domain of sequential data, three synthetic datasets have been implemented to determine the performance of a building block at three levels of difficulty. The validation accuracy thresholds for these datasets are set to 80%, 70%, and 40%, respectively. In the following, building blocks that surpass all three thresholds are referred to as ELENA, short for elementary neural architecture.

The first and most simple **Reber grammar dataset** describes a one-class classification task, that involves learning a generated sequence of symbols that follow the rules of the Reber grammar. It serves as a time-inexpensive check on whether a building block is capable of learning a relatively simple structure in the data, namely short-range dependencies. It was chosen as the first task since it exhibits properties that are difficult for traditional machine learning algorithms but usually rather simple for recurrent neural networks (RNN). The second **embedded Reber grammar dataset** is an

extension of the Reber grammar dataset and generates string sequences with extended time lags. This task aims to check the ability of the building block to learn longer-range dependencies. The third and most complex **adding problem dataset** describes a regression task where two random values in a long sequence have to be summed. The sequence is assumed to be processed correctly if the absolute distance of the prediction to the target is below 0.04. This task determines the ability of the building block to solve long time-lag problems involving continuous-valued representations. To test the performance of the best building blocks found during the search procedures, a fourth **memory dataset** has been implemented. Unlike the other three datasets, it concerns a multi-class classification task where the input at the initial time step has to be memorized for a predefined sequence length. It is not used in the search process, but used to rank the resulting building blocks.

## 4.2 Building block search

In total, four search processes were conducted for approximately a thousand episodes with the final hyperparameters illustrated in Section F of the appendix. Each of the search processes resulted in at least one ELENA, indicating that the success is not contingent on a specific random weight initialization. Out of the seven ELENAs found during the four search processes, we detail the two building blocks with the highest performance, named ELENA1 and ELENA2. The performance was evaluated on the unseen memory task with various sequence lengths as detailed in Section G of the appendix.

**ELENA1**. This building block utilizes two weight matrices to transform the input used to compute the single hidden state as well as the output. The hidden state $\mathbf{h}_1(t)$ gets activated by the tanh activation function. The output of the previous time step $\mathbf{y}(t-1)$ is used to compute the output of the current time step $\mathbf{y}(t)$:

$$
\begin{aligned}
\mathbf{h}_1(t) &= \tanh\left(\mathbf{W}_1\mathbf{x}(t) + \mathbf{b}_1\right) \\
\mathbf{y}(t) &= \left(\mathbf{W}_2\mathbf{x}(t)\right) \odot \mathbf{h}_1(t) + \mathbf{y}(t-1)
\end{aligned}
\tag{2}
$$

**ELENA2**. This building block also utilizes two weight matrices to transform the input. The hidden state of the previous time step $\mathbf{h}_1(t-1)$ is added to the hidden state of the current time step $\mathbf{h}_1(t)$ before it is activated by the *tanh* activation function, which follows the concept of the cell state of the LSTM (Hochreiter and Schmidhuber, 1997). Furthermore, $\mathbf{h}_1(t)$ is then element-wise multiplied by the transformed input $\mathbf{W}_2\mathbf{x}(t)$ to compute the output $\mathbf{y}(t)$, which follows a similar logic as the output gate, similar to ELENA1. Also, the output of the previous time step $\mathbf{y}(t-1)$ is added to the output of the current time step $\mathbf{y}(t)$:

$$
\begin{aligned}
\mathbf{h}_1(t) &= \tanh\left(\mathbf{W}_1\mathbf{x}(t) + \mathbf{h}_1(t-1) + \mathbf{b}_1 + \mathbf{b}_2\right) \\
\mathbf{y}(t) &= \left(\mathbf{W}_2\mathbf{x}(t)\right) \odot \mathbf{h}_1(t) + \mathbf{b}_2 + \mathbf{y}(t-1)
\end{aligned}
\tag{3}
$$

**Performance evaluation on real-world datasets**. To assess whether the developed architectures ELENA1 and ELENA2 can be transferred to complex real-world tasks, we first applied them to the well-established, publicly available Tox21 dataset (Mayr et al., 2016). The dataset comprises 12 different prediction tasks corresponding to distinct toxic effects. The inputs are small molecules whose chemical structure is represented as a sequence, in the so-called Simplified Molecular Input Line Entry System (SMILES) notation (Weininger, 1988). It represents a difficult multi-task sequence classification problem, in which high-level features, such as chemical reactivity, have to be extracted from a low-level representation of the data. To conduct the experiments, a single recurrent building block can be stacked to form layers of the overall neural network. For fair comparison, the number of stacked layers, together with the batch size and the learning rate, were optimized on the validation set. The average test and validation ROC-AUC together with the corresponding standard deviation over ten individual runs with the original splits of the Tox21 Data Challenge are reported in Table 1.

Table 1: Predictive performance on the Tox21 and PTB datasets. ROC-AUC represents the average area under receiver operating characteristic curve, with higher values indicating better classification performance across 12 tasks. Perplexity measures the average confidence on a word-level, with lower values indicating better language modeling performance across 10,000 words.

| Model | Tox21 dataset | | | PTB dataset | | |
|---|---|---|---|---|---|---|
| | Param. | Test ROC-AUC | Val ROC-AUC | Param. | Test Perplexity | Val Perplexity |
| GRU | 0.68M | 78.52 ± 1.06 | 77.20 ± 0.79 | 8.28M | 195.28 ± 2.36 | 214.13 ± 2.54 |
| LSTM | 0.84M | 77.35 ± 1.06 | 75.27 ± 1.73 | 7.23M | 124.51 ± 1.67 | 131.04 ± 2.11 |
| ELENA1 (ours) | 0.34M | 77,33 ± 0.81 | 76.54 ± 0.82 | 6.18M | 218.53 ± 2.38 | 228.04 ± 2.68 |
| ELENA2 (ours) | 0.17M | 76.70 ± 0.58 | 77.49 ± 0.60 | 6.18M | 185.67 ± 2.00 | 195.38 ± 2.41 |
| NAS2 | 0.38M | 76.61 ± 0.74 | 75.32 ± 1.06 | 9.33M | 230.27 ± 2.42 | 249.13 ± 3.08 |
| ENAS | 0.58M | 75.83 ± 1.04 | 73.63 ± 1.35 | 11.95M | 122.40 ± 1.54 | 130.04 ± 1.61 |
| DARTS | 0.46M | 74.59 ± 0.60 | 73.42 ± 0.94 | 10.37M | 142.44 ± 1.60 | 151.29 ± 1.97 |
| NAS1 | 0.38M | 73.38 ± 1.69 | 73.98 ± 0.87 | 9.33M | 242.35 ± 2.72 | 266.39 ± 4.03 |
| RNN | 0.51M | 73.13 ± 1.01 | 69.29 ± 1.06 | 6.19M | 449.08 ± 4.76 | 499.65 ± 6.52 |

Second, we applied ELENA1 and ELENA2 to the well-established Penn Tree Bank (PTB) dataset (Marcus et al., 1993), a widely used benchmark for evaluating models on natural language processing capabilities, assessing the understanding of syntactic and semantic patterns in a large collection of text. The primary goal of the task is next word prediction, namely predicting the next token in a sequence of tokens. We estimated the performance using bootstrapping, thus evaluated the mean and standard deviation of the validation and test perplexity on 100 randomly drawn subsets with replacement.

Besides comparing our building blocks to the benchmark LSTM (Hochreiter and Schmidhuber, 1997), the Gated recurrent unit (GRU) (Chung et al., 2014), and the Elman network (RNN) (Elman, 1990), we also compare them to the neural architectures that resulted from other NAS approaches, namely NAS1 and NAS2 (Zoph and Le, 2016), ENAS (Pham et al., 2018) and DARTS (Liu et al., 2018). It is important to acknowledge that, due to computational limitations, the overall model sizes and hyperparameter configurations regarding the PTB task were constrained, leading to an comparatively increased perplexity (Zaremba et al., 2014).

ELENA1 and ELENA2 are on par with or even outperform the other architectures on the Tox21 task, while only being about 40% to 90% (ELENA1) and 20% to 45% (ELENA2) their size. Moreover, even though the NAS architectures that we compared against were explicitly searched on the PTB dataset, ELENA1 and ELENA2 demonstrate substantial performance on the PTB task with only a fraction of their size. This confirms that ELENAs are parameter-efficient and show good generalization across tasks.

## 5 Conclusion

We demonstrated that Elementary Neural Architecture Search (ELENAS) is feasible and leads to high-performing building blocks that transfer well to real-world data. The general search space of elementary mathematical operations allows for high flexibility and includes unprecedented building blocks, such as the presented ELENA1 and ELENA2 for the domain of sequential data. They are to be easily transferred to a wide variety of problems as they generalize well and can be scaled in terms of computational cost and learnable parameters. We showed that it is worth further exploring the general search space of elementary mathematical operations to find unprecedented general-purpose building blocks.

**Limitations**. The conducted experiments of our work were restricted to a relatively narrow cell-based search space and limited evaluation of the building blocks. Further work could expand this search space to a wider selection of elementary mathematical operations. The macro connections among the building blocks may be analyzed in future work as well. Last but not least, our experiments were limited to the domain of sequential data, but since our method is applicable in multiple domains, a search for novel building blocks in other domains should be exploited.

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

## A  Broader Impact Statement

ELENAS is capable of designing computationally efficient neural architectural building blocks that require only a fraction of learnable parameters compared to state-of-the-art neural networks while achieving similar performance. Also, although Neural Architecture Search usually comes with a significant environmental cost, our method proposes an evaluation procedure that is, compared to the majority of other NAS approaches, computationally inexpensive. Thus, our method supports reducing the carbon footprint when searching and subsequently deploying ELENAs. After careful reflection, we have determined that this work presents no notable negative impacts to society or the environment.


## C  GNN message passing-layers

**Acknowledgements**. To construct the GNN that processes the state of the RL setting, we chose the message-passing layers *HEATConv*, short for heterogeneous edge-featured graph attention network, proposed by Mo et al. (2021). These layers are an extension of the Graph Attention Network (GAT) by Veličković et al. (2017), and achieved the highest performance on an artificially created self-supervised task among six message-passing layers that all incorporate edge features (Mo et al., 2021; Veličković et al., 2017; Shi et al., 2020b; Brody et al., 2021; Rozemberczki et al., 2020; Li et al., 2020). The ablation study concerned the prediction of all edge types of various computation graphs, where the data was collected by simulating episodes that generate state-of-the-art recurrent neural architectures, such as variants of the LSTM architecture (Hochreiter and Schmidhuber, 1997). One of the success factors of *HEATConv* is that – additionally to node and edge features – it supports node and edge types. In the present work, the first node type is assigned to *input* and *initial* nodes, while the second node type is assigned to all other nodes. The first edge type is assigned to all edges, while the second edge type is assigned to *backwards* edges that are not being predicted but only improve the information flow in the graph.

## D  Neural constructor

The neural constructor converts a neural architectural building block to a neural architecture that can be trained and evaluated. When constructing the neural architecture, some adjustments are made to better avoid computation graphs that cannot be evaluated:

(i) Recursions to the *input* node are not considered, as the input should not be overwritten.

(ii) *Intermediate* nodes that only have incoming edges of type *none* are not considered as part of the computation graph but are considered *dead* nodes as they cannot be computed.

(iii) *Final* nodes that have no direct or indirect connection to the *input* node are considered *dead* nodes, since they are not processing the input.

(iv) In the case of multiple *final* nodes, the node that has the most parents is considered *final*, while the others are considered *dead*, since only one *final* node can be computed. Thereby, we assume that the *final* node that has the highest number of parents commonly represents the neural architecture with the highest complexity.

(v) Nodes that are used in some computations – meaning nodes that have incoming or outgoing connections of type other than *none* – but that are not parents of the selected *final* node are considered *dead*, since they do not contribute to the computation of the *final* node.

The same neural architecture can be evaluated multiple times on different datasets of varying difficulty to get an accurate performance estimate that generalizes to the whole domain. Depending on whether the dataset at hand concerns a classification or a regression task, the loss function is defined as the cross-entropy loss or mean squared error loss, respectively (Goodfellow et al., 2016). In case the *final* node is assigned with a specific activation function, it is removed, since the implementation of all loss functions – namely, mean squared error loss for regression tasks, binary cross entropy loss for one-class classification tasks, and categorical cross-entropy loss for multi-class classification tasks (Goodfellow et al., 2016) – requires raw non-activated predictions (also called logits) as input, since they already incorporate the applicable activation function, if any (Paszke et al., 2019).

To initialize all weights and biases represented by *initial* nodes, their dimensions have to be evaluated. Therefore, each operation has a dedicated function that attempts to find the correct dimensions of the tensors used for the respective operation. Once all dimensions are evaluated, a sanity check is performed to ensure that all dimensions fit together and match the given input dimension as well as the desired output dimension. In case the dimensions are valid, *initial* nodes are randomly initialized as parameters that require the computation of gradients, according to a normal distribution with a mean of zero and standard deviation of 0.1 (Paszke et al., 2019). Recursion nodes are initialized as zeros tensors. Also, a training and validation data loader is initialized (Paszke et al., 2019). Thereafter, for each time step in the sequence, all *intermediate* nodes are computed according to a computation order. The actual computation is performed by calling respective functions related to the operation. The computed *final* node is then the output of the neural architecture, which is used to compute the loss and propagate back the errors using gradient descent (Goodfellow et al., 2016). Once the neural architecture is trained, the validation accuracy is determined on the validation data.

Once a new operation is to be used in the search space, the only thing that has to be defined is the respective function that determines the dimensionality of the node used in the operation as well as a respective function that actually performs the computation. In this way, any arbitrary operation can be added to the search algorithm, enabling high flexibility for different kinds of search spaces.

Table 2: Reward shaping as part of the overall reward function

| Category | Description | Frequency | Δ Reward | Impact | Value $r$ |
|---|---|---|---|---|---|
| Action | input node with outgoing edges of type *not_present* only | every step | $r_{a_1}$ | negative | 0 |
| | more than $i$ incoming edges $e$ of type *matmul* | if $i$ increases | $(e - i) * r_{a_3}$ | negative | 0.001 |
| | more than $i$ incoming edges $e$ of type *hadamard* | if $i$ increases | $(e - i) * r_{a_4}$ | negative | 0.001 |
| | more than $i$ incoming edges $e$ of type *add* | if $i$ increases | $(e - i) * r_{a_5}$ | negative | 0.001 |
| | more than $i$ incoming edges $e$ of type *recursion* | if $i$ increases | $(e - i) * r_{a_6}$ | negative | 0.001 |
| Node | initial nodes $n$ | if $n$ increases | $n * r_{n_3}$ | positive | 0 |
| | intermediate nodes $n$ | if $n$ increases | $n * r_{n_4}$ | positive | 0 |
| | dead nodes $n$ | if $n$ increases | $n * r_{n_5}$ | negative | 0 |
| | unused nodes $n$ | if $n$ increases | $n * r_{n_6}$ | negative | 0 |
| | no initial node | every step | $r_{n_7}$ | negative | 0 |
| | no recursion node | every step | $r_{n_8}$ | negative | 0 |
| | no final node | every step | $r_{n_9}$ | negative | 0 |
| Graph | illegitimate recursion present | every step | $r_{g_1}$ | negative | 0 |
| | dimensions do not fit together | every step | $r_{g_2}$ | negative | 0 |
| | architecture can successfully be trained, but $a_{val} < a_{req}$ | every step | $r_{g_3}$ | positive | 0.01 |
| | no proper architecture found | end of episode | $r_{g_4}$ | negative | 0 |

## E Reward shaping

The reward is shaped by inspecting the computation graph to enforce desired behavior. The additional reward shaping can be adjusted according to table 2, with $r_{a_i}, r_{n_i}, r_{g_i} \in \mathbb{R}, i \in \mathbb{N}$. The category *Action* concerns the types of all incoming or outgoing edges of each node, the category *Node* concerns the types of all nodes within the graph, and the category *Graph* refers to the overall structure and properties of the building block.

## F Hyperparameter

All hyperparameters that specify the training procedure are categorized into eight different categories. First, *Graph* concerns the structure of the computation graph. Second, *GNN* specifies the overall neural network structure and usage of the GNN. Third, *Actor* specifies the GNN-based neural network structure of the actor. Fourth, *Critic* specifies the neural network of the critic including its graph global pooling operator. Fifth, *Designer* concerns the actual PPO algorithm Schulman et al. (2017). Sixth, *Designee* defines the construction and evaluation of the building block. Seventh, *Dataset* defines the datasets the building block is evaluated on as well as their relative importance. Last but not least, *Logging* deals with the extent of logging the process as well as the results of ELENAS. The self-supervised task described in Section has been utilized to determine the best hyperparameters of both the graph representation as well as the PPO algorithm. A complete list, including a brief description of each hyperparameter can be found in table 4.

**Number of total nodes in** $\mathbb{G}$. Opting for a total of 32 nodes results in one input, 16 initial and 15 intermediate nodes. This computation graph $\mathbb{G}$ comprises 480 edges, with each edge type predicted once to construct a building block. Such a $\mathbb{G}$ can potentially represent most state-of-the-art building blocks. Moreover, having 15 intermediate nodes proves advantageous for dividing $\mathbb{G}$ into three subgraphs, to deterimne the order in which the edges are predicted, as detailed below. Additionally, since the number of edges increases quadratically with the number of nodes, choosing a total of 32 nodes provides a suitable balance between the complexity of the graph structure and computational expenses. However, instead of fixing the number of total nodes in $\mathbb{G}$, initiating the search process with a relatively small number and incrementally increasing it could potentially lead to enhanced computational efficiency and search performance, providing an avenue for future research.

**Positional encoding of node and edge features**. To study the impact of the positional encodings of both the node as well as the edge features, a GNN based on *HEATConv* (Mo et al., 2021) layers and a subsequent multilayer perceptron (MLP) (Goodfellow et al., 2016) has been applied on the self-supervised task described in Section with different edge and node feature combinations. First, everything else equal, the accuracy of predicting the correct edge types increases when including the positional encoding as part of the edge features. Second, everything else equal, the accuracy of predicting the correct edge types was the highest for positional encoding as part of the node features when constructed using a trigonometric function, namely the sine function, compared to using equally distributed values between 0 and 1, or not using positional encodings at all. Furthermore, it could empirically be observed that this additional information improved the overall search process.

**Sub-graphs of $\mathbb{G}$ to determine edge prediction order**. The order in which edge types are predicted as part of the search process can be chosen arbitrarily. Thus, we studied the impact of different orderings on the search performance. One straightforward ordering involves predicting all edges related to a single node before moving on to the next node. However, we empirically concluded that splitting $\mathbb{G}$ into sub-graphs and predicting edges within each sub-graph in a sequential manner, facilitates the discovery of high-performing building blocks. To be precise, for a total of 32 nodes in $\mathbb{G}$, we chose five intermediate nodes to form a sub-graph, resulting in three sub-graphs. Since node $n_i$ is defined as intermediate in case $i$ is even, intermediate nodes $n_i$ with $i \in \{2, 4, 6, 8, 10\}$ constitute the first sub-graph. All edges that connect the input node $n_0$ and the initial nodes with the intermediate nodes of this sub-graph are predicted first (including self-connections of these intermediate nodes). The second sub-graph includes intermediate nodes $n_i$ with $i \in \{12, 14, 16, 18, 20\}$. Similarly, all edges that connect the input node $n_0$ and the initial nodes with the intermediate nodes of this sub-graph are subsequently predicted (including self-connections of these intermediate nodes). This process is repeated for all intermediate nodes in $\mathbb{G}$, determining the order in which edges are predicted during the search process.

# G Performance evaluation on synthetic dataset

Each of the building blocks that achieved a validation accuracy of over 80% on the Reber grammar dataset, 70% on the embedded Reber grammar dataset, and 40% on the adding problem dataset, has been further evaluated on the memory task, which has never been used during the search process. Therefore, we randomly initialized its learnable parameters to train each building block from scratch. We report the average test accuracy [%] and the standard deviation over the ten individual runs in table 3. It can be observed that the two highest performing building blocks, ELENA1 and ELENA2, outperform the benchmark building block, the LSTM (Hochreiter and Schmidhuber, 1997), in all but one sequence length setting. The numbering of ELENAs has been chosen arbitrarily.

Table 3: Predictive performance on the unseen synthetic memory dataset in terms of test accuracy across various sequence lengths. Higher performance achieved by ELENA1 or ELENA1 compared to LSTM (Hochreiter and Schmidhuber, 1997), is displayed in bold.

| Building Block | Sequence Length | | | | | | |
|---|---|---|---|---|---|---|---|
| | 20 | 30 | 40 | 50 | 60 | 70 | 80 |
| **LSTM** | $73.08 \pm 20.15$ | $49.18 \pm 25.42$ | $24.07 \pm 11.43$ | $20.13 \pm 6.60$ | $26.82 \pm 24.01$ | $24.17 \pm 26.85$ | $40.63 \pm 40.18$ |
| **ELENA1** | $57.06 \pm 1.01$ | $\mathbf{57.11 \pm 0.51}$ | $\mathbf{56.88 \pm 0.66}$ | $\mathbf{57.22 \pm 0.87}$ | $\mathbf{56.89 \pm 0.86}$ | $\mathbf{56.97 \pm 0.65}$ | $\mathbf{56.72 \pm 2.20}$ |
| **ELENA2** | $\mathbf{99.83 \pm 0.11}$ | $\mathbf{98.80 \pm 0.35}$ | $\mathbf{96.34 \pm 1.04}$ | $\mathbf{90.64 \pm 3.37}$ | $\mathbf{89.41 \pm 3.09}$ | $\mathbf{83.14 \pm 4.20}$ | $\mathbf{74.01 \pm 7.17}$ |
| **ELENA3** | $57.26 \pm 0.81$ | $\mathbf{57.03 \pm 0.75}$ | $\mathbf{57.03 \pm 1.42}$ | $\mathbf{55.93 \pm 2.62}$ | $\mathbf{47.10 \pm 8.88}$ | $\mathbf{41.52 \pm 14.08}$ | $\mathbf{41.30 \pm 9.68}$ |
| **ELENA4** | $55.94 \pm 1.52$ | $\mathbf{55.05 \pm 2.91}$ | $\mathbf{53.13 \pm 3.06}$ | $\mathbf{50.65 \pm 4.60}$ | $\mathbf{49.32 \pm 4.84}$ | $\mathbf{50.03 \pm 4.42}$ | $\mathbf{50.95 \pm 4.08}$ |
| **ELENA5** | $54.79 \pm 2.96$ | $\mathbf{51.45 \pm 6.23}$ | $\mathbf{49.16 \pm 7.54}$ | $\mathbf{51.93 \pm 3.68}$ | $\mathbf{50.22 \pm 4.85}$ | $\mathbf{49.68 \pm 4.90}$ | $\mathbf{48.15 \pm 7.35}$ |
| **ELENA6** | $57.14 \pm 1.33$ | $\mathbf{54.32 \pm 5.31}$ | $\mathbf{50.16 \pm 7.68}$ | $\mathbf{43.86 \pm 8.69}$ | $\mathbf{29.49 \pm 4.80}$ | $\mathbf{26.12 \pm 8.28}$ | $20.98 \pm 7.29$ |
| **ELENA7** | $42.81 \pm 9.49$ | $38.61 \pm 8.64$ | $\mathbf{35.56 \pm 9.09}$ | $\mathbf{32.02 \pm 5.78}$ | $\mathbf{27.32 \pm 9.34}$ | $23.46 \pm 7.75$ | $19.01 \pm 5.74$ |

Table 4: Hyperparameters regarding the ELENAS methodological framework

| Category | Hyperparameter | Description | Type | Default Value |
|---|---|---|---|---|
| Graph | combining_operations | potential operations used to compute intermediate/final nodes | list of strings | ['matmul', 'had', 'add', 'rec'] |
| | transforming_operations | potential operations used to transform intermediate/final nodes | list of strings | ['sigm', 'tanh', 'relu', 'trans'] |
| | allow_recursion | whether recursive connections are allowed | bool | True |
| | number_total_nodes | number of nodes in the computation graph | $\mathbb{N}$ | 32 |
| GNN | graph_layer_type | message passing layer used by the GNN | string | 'HEATConv' |
| | common_backbone | whether to use a common GNN backbone for actor and critic | bool | True |
| | number_gnn_layers | number of message passing layers | $\mathbb{N}$ | 2 |
| | number_heads | number of heads of each message passing layer | $\mathbb{N}$ | 4 |
| | emb_concat | whether to concatenate multi-head attentions | bool | False |
| | node_emb_dimension | node embedding dimension after message passing steps | $\mathbb{N}$ | 256 |
| | edge_emb_dimension | edge embedding dimension during message passing steps | $\mathbb{N}$ | 256 |
| | leakyrelu_negative_slope | LeakyReLU angle of the negative slope | $\mathbb{R}$ | 0.01 |
| | graph_dropout | dropout probability during GNN training | $\mathbb{R} \in [0, 1]$ | 0 |
| | load_pretrained_gnn | whether to load a pre-trained GNN | bool | False |
| Actor | actor_number_linear_layers | depth of fully-connected neural network of actor | $\mathbb{N}$ | 3 |
| | actor_reduce_last_layer_by | factor by which the number of last hidden layer neurons is reduced | $\mathbb{N}$ | 4 |
| Critic | pooling_layer_type | global pooling layers used to compute graph embedding | string | 'GMT' |
| | gmt_number_heads | number of heads for global pooling layers (GMT layer specific) | $\mathbb{N}$ | 4 |
| | gmt_use_self_att | whether to use the self-attention layer | bool | False |
| | critic_number_linear_layers | depth of fully-connected neural network of critic | $\mathbb{N}$ | 2 |
| | critic_reduce_last_layer_by | factor by which the number of last hidden layer neurons is reduced | $\mathbb{N}$ | 1 |
| PPO | ppo_epochs | number of epochs the RL agent is trained within each update | $\mathbb{N}$ | 4 |
| | ppo_batch_size | number of samples in a batch used to update the actor-critic network | $\mathbb{N}$ | 32 |
| | ppo_buffer_capacity | maximum number of samples stored in the replay buffer | $\mathbb{N}$ | 9600 |
| | ppo_lr | learning rate for updating the RL agent | $\mathbb{R}$ | 0.001 |
| | ppo_betas | coefficients for computing running averages of gradients | tuple | (0.9, 0.999) |
| | ppo_weight_decay | factor by which weights of Designer are penalized for regularization | $\mathbb{R}$ | 0.0001 |
| | ppo_discount | factor by which future reward is discounted | $\mathbb{R}$ | 0.99 |
| | ppo_episodes | number of total episodes sampled during a run | $\mathbb{N}$ | 1000000 |
| | ppo_loss_scales | weight of value loss, policy loss, and entropy loss to assess total loss | 3-tuple | (0.5, 1.0, 0.01) |
| | ppo_epsilon_clip | value to set the clip interval on the probability ratio term $[1 - \epsilon, 1 + \epsilon]$ | $\mathbb{R}$ | 0.2 |
| | load_agent | whether to load a trained RL agent | bool | False |
| | ppo_shuffle_edges | whether to have a random edge order during each episode | bool | False |
| | ppo_subgraph_size | number of intermediate nodes in subgraphs to determine edge order | $\mathbb{N}$ | 5 |
| Designee | designee_hidden_features | number of neurons in the hidden dimension | $\mathbb{N}$ | 64 |
| | designee_epochs | number of epochs building block is trained before determining the accuracy | $\mathbb{N}$ | 1 |
| | designee_batch_size | number of samples in a batch used to update building block | $\mathbb{N}$ | 32 |
| | designee_optimizer | which algorithm to use for updating the weights | string | 'Adam' |
| | designee_learning_rate | learning rate for updating the building block | $\mathbb{R}$ | 0.001 |
| | designee_weight_decay | factor by which weights are penalized for regularization | $\mathbb{R}$ | 0 |
| Dataset | datasets | datasets used to evaluate the performance of building block | list of strings | ['reb', 'erg', 'add'] |
| | datasets_thresholds | accuracy threshold for declaring a building block as proper | tuple | (0.8, 0.7, 0.4) |
| | datasets_avg_accuracy | average validation accuracy that untrained architectures achieve | tuple | (0.5, 0.5, 0.15) |
| | datasets_reward_scale | factor by which the reward is scaled for each dataset | tuple | (0.1, 0.1, 1) |
| | datasets_sequence_lengths | maximum number of time steps per sample of each dataset | tuple | (32, 32, 32) |
| | datasets_number_samples | number of training samples per dataset (+10% validation data) | tuple | (20k, 50k, 90k) |
| Logging | log_path | path where log files are stored | os.path object | - |
| | log_architectures | whether to log architectures achieving accuracies above the threshold | bool | True |
| | log_exceptions | whether to log exceptions that occur while running the search process | bool | True |
| | log_graph_visualizations | whether to log computation graph visualizations as .html files | bool | True |

