# OpenReview forum: "ELENAS: Elementary Neural Architecture Search"
_automl.cc/AutoML/2023/Conference — AutoML 2023 Workshop_

### Official Review · Reviewer_s9Zr · 2023-04-12

**Potential Impact On The Field Of Automl Rating:** 3
**Technical Quality And Correctness Rating:** 3
**Clarity Rating:** 3

**Summary Of Contributions:**

This paper proposes ELENAS, which is a method that searches for mathematical operations in a newly defined search space to form building blocks, i.e., combinations of operations applied on input features similar to one operation in a cell-based search space. This method forms these building blocks as computational graphs and utilizes Graph Neural Networks to learn node and graph embeddings. These embeddings are part of the search strategy, here proximal policy optimization. This paper shows that the proposed method is able to find efficient novel building blocks.

**Actions Required To Increase Overall Recommendation:**

In order to increase the overall recommendation, this paper would need to answer the open questions in Technical Quality And Correctness.

**After Rebuttal**: I keep my score of 5.

**Clarity:**

In the introduction, this paper is not able to clearly state what a building block is. From the NAS perspective, it could be interpreted as new cells, instead of the actual operation building block.

Lines: 41-43 not clear what is meant here

Line 156: why is m=32, especially if it’s better to split the computational Graph G into subgraphs eventually?

Figure 2: the edge feature matrix is not clear to me and can be misleading.  It is not clear, when the nodes are combined and when transformed.

L: 184-185:  Why does the number of transforming operations must be equal to the number of combination operations?

Eq. 1 / line 158: no convolution operation, but a multiplication.


**Overall Review:**

Strengths:
The experiments are well done and explore an interesting idea of new building blocks, reducing the amount of human-expert knowledge.
This paper is well-written and motivates the idea good.

Weaknesses:
There are some open questions (see Technical Quality And Correctness), which would when answered improve the overall assessment of the approach of finding novel and efficient building blocks.
My main concern is that there isn’t any standard benchmark from the NAS literature used and the stacking is not mentioned clearly.

Missing citation, which also introduces different building blocks.
Nicholas Roberts, Mikhail Khodak, Tri Dao, Liam Li, Christopher Ré, Ameet Talwalkar:
Rethinking Neural Operations for Diverse Tasks. NeurIPS 2021


**Potential Impact On The Field Of Automl:**

This work introduces an interesting approach, which could be eventually used for AutoML if further evaluated.

**Review Confidence:**

3: You are fairly confident in your assessment. It is possible that you did not understand some parts of the submission or that you are unfamiliar with some pieces of related work.

**Review Rating:**

5: Borderline Leaning Reject: Technically sound paper where reasons to reject nonetheless outweigh reasons to accept. Please use sparingly.

**Review Summary:**

While the idea of searching for novel building blocks using GNNs with RL is novel,
I believe additional assessments and experiments are needed to improve this paper and clearly show its superiority over other search approaches.


**Technical Quality And Correctness:**

There are some open questions, with would support the proposed approach regarding its technical quality and ability to improve over other methods.

RL and PPO are seldom used in NAS in recent research. Therefore, as the authors also mention, other approaches could be used, e.g., Bayesian optimization. Thus, how could the proposed method be adapted to the BO setting and what would be the disadvantage over the used RL setting in this paper? A discussion on that would be helpful.

The used stacking of building blocks, and thus the “macro” architecture is not mentioned in this paper.
This leads to the question if the boost in Table 2 comes from ELENA1/ELENA2 or only the stacking approach.

In addition, this paper does not compare to known benchmarks and datasets such as Penn TreeBank, which would eventually better assess the ability of the proposed method.

How were all hyperparameters set in this paper? Especially also for the shaping reward r_shape in Table  3 (0.001, 0.01)?

What are the search times?
How is the efficiency of the building blocks elaborated and estimated (line 16)?

---

> ### Author Response · Authors · 2023-05-02
> **Author response to Reviewer s9Zr**
>
> We appreciate your constructive feedback on our work and recognize the concerns you raised. To address them, we have made the following changes in our manuscript:
>
> 1. Search Strategy: We have extended our discussion on the choice of search strategy. In general, we opted for RL learning, since a RL agent can acquire knowledge from graph structures to consequently refine the search space of elementary mathematical operations that only includes a relatively low fraction of high-performing building blocks.
>
> 2. Stacking of Building Blocks: We have provided a clearer explanation of the stacking approach in Section F of the appendix. In essence, stacking simply refers to having multiple layers of building blocks of the same type to construct the overall model to increase the model complexity. Thus, the boost can still be attributed to the individual building blocks, since everything else is equal for each of the compared models.
>
> 3. Hyperparameter Settings: We have also provided a more detailed explanation of the final hyperparameter setting. Regarding the reward shaping, the values r in Table 2 of the revised paper are the ones used in the respective formula for the final reward function detailed in Section 3.4.
>
> 4. Results on additional real-world datasets: As per your suggestion, we have included results on the Penn TreeBank dataset to provide a better assessment of our method's capabilities in finding parameter-efficient and well-generalizable building building blocks.
>
> 5. Search Times and Efficiency: In general, each search process requires about 100 GPU hours on a single NVIDIA A100 GPU. Since we conducted four separate search processes, we reported a total of 400 GPU hours in our OpenReview submission. However, ELENA1 was already found at about 52% and ELENA2 was found at about 41% of the search process. Also, the efficiency of the building blocks is measured by the number of learnable parameters.
>
> 6. Number of nodes 𝑚 and graph-splitting:  We have provided more insights into the potential implications of the number of nodes 𝑚 in the computation graph. In general, we have opted for 32 nodes, since such a computation graph can potentially represent most state-of-the-art building blocks. Additionally, since the number of edges increases quadratically with the number of nodes (formula provided in Section 3.1), it provides a suitable balance between the complexity of the graph structure and computational expenses. Also, it proves advantageous to split the graph into 3 subgraphs to determine the order in which edge types are predicted. We have added more information on this regard in Section F of the appendix. In essence, this approach simply refers to the order in which the edge types are predicted, without modifying the graph itself.
>
> 7. Clarity-related Issues:
> - Although there is no formal definition of building blocks, as part of the abstract they are exemplified as components of deep learning models, such as feed-forward, convolution, recurrent, long short-term memory, or attention blocks. Also, especially for the domain of computer vision, building blocks are exemplified as normalization layers, pooling layers, and convolution layers with different filter sizes. This should give a clear understanding of what we refer to as building blocks.
> - Line 41-43: “In contrast to well-established search spaces in NAS, the majority of building blocks included in this search space of elementary operations are not valid and thus cannot be used in neural networks.” This statement indicates that our search space encompasses a large number of building blocks that do not represent a proper neural network that can be trained on data, since the elementary mathematical operations do not fit together. In contrast, established search spaces within NAS solely consist of trainable neural networks, where only certain attributes such as the kernel size are altered.
> - Figure 2: As indicated in Section 3.2, combining operations are for edges that connect a node with a different node, while transforming operations are for edges that connect a node with itself. This reduces the search space by a factor of two without compromising its generality. Since the number of edge types is fixed, the number of transforming operations theoretically must be equal to the number of combination operations (practically, in case more combining operations are desired, additional transforming operations can be defined as the identity that has no effect on the node)
> - We have added the missing citation (Roberts et al., NeurIPS 2021) as suggested.
>
> We hope that these revisions address your concerns and provide a clearer and more comprehensive understanding of our work, encouraging you to reconsider your assessment. Thank you once again for your valuable feedback!

---

> > ### Comment · Reviewer_s9Zr · 2023-05-08
> > **Author Response**
> >
> > I appreciate the author's response and will update the *Potential Impact On The Field Of AutoML Rating* and the *Technical Quality And Correctness Rating*. The PTB numbers however show limited improvements to other search methods, while using rather huge computational costs. Therefore I agree with Reviewer LpEh about the uncertainty of how the proposed method scales to larger datasets and thus will keep my score.

---

> > > ### Author Response · Authors · 2023-05-08
> > > **Author response**
> > >
> > > Thank you very much for the response. As already mentioned to Reviewer LpEh, the novelty of our method lies in its ability to find general-purpose building blocks by solely using synthetic datasets. As demonstrated in the real-world experiments, these building blocks can then be scaled to various datasets of any size and complexity while being very parameter-efficient. Thus, our method has already reached its maximum computational complexity, requiring as little as 2 GPU days to find high-performing ELENAs. In comparison, Efficient NAS (ENAS) finds recurrent cells in about 0.5 GPU days, DARTS in about 1 GPU day, and NAS in about 10,000 CPU days (Liu et al., 2018). In light of this clarification, we would be highly appreciative if you could re-evaluate your overall review rating as well. Thanks!

---

### Official Review · Reviewer_NxY5 · 2023-04-13

**Potential Impact On The Field Of Automl Rating:** 3
**Technical Quality And Correctness Rating:** 3
**Clarity Rating:** 3

**Summary Of Contributions:**

The paper presents ELENAS, a method for finding novel, performant neural architectural building blocks at the elementary level by combining mathematical operations. The building blocks are represented as computational graphs and processed by GNNs as part of a reinforcement learning system with the Proximal Policy Optimization (PPO) algorithm. Ultimately, ELENAS can design building blocks and is employed in the domain of sequential data. ELENA1 and ELENA2 building blocks are proposed for the sequential data domain with better performance compared to LSTM. The performance metric is based on the proposed reward function based on the average accuracy, a specific step, the factor by which the reward is weighted for a specific dataset, and reward shaping. The resulting building blocks are compared based on the accuracy of the test/validation set.

**Actions Required To Increase Overall Recommendation:**

Address the minor issues: Ensure that minor issues such as the reference to the correct table and the explanation of the ROC-AUC performance metric are corrected and clarified, which would improve the overall readability and understanding of the paper.

Enhance the results in Table 1: As suggested earlier, either improve Table 1 by including results for all four datasets and comparisons with other building blocks besides LSTM, or provide a clear explanation for focusing on only one dataset in the experiments and the rationale behind comparing the method only with LSTM.

**Clarity:**

There are some minor concerns that can be addressed to improve the paper quality:
- The authors provide the details of the process of designing a building block and the use of the PPO algorithm, but there is no discussion on the advantages of PPO over other potential RL algorithms, which would help establish the reason for this choice.
- The discussion on the size of G and the number of total nodes 𝑚 is clear, and the authors mention that it’s possible to increase 𝑚 to adapt to the search progress dynamically. However, they could provide more insight into the potential implications of increasing 𝑚, such as the impact on computational complexity and search efficiency. This would help to understand the trade-offs in adjusting 𝑚.
- In line 314, the sentence mistakenly refers to Table 1, while it should actually refer to Table 2.
- The ROC-AUC performance metric in Table 2 which is presumably the area under the ROC, is not explained in the paper or the table caption. A brief explanation of what it means would help the readers understand whether a higher or lower value is better.
- In AutoML paper format, the captions for tables should be placed at the top. However, in this paper, the captions for both tables are located below the tables.
- The methodology section presents a search space containing elementary mathematical operations assigned to edge types. The authors propose reducing the search space by a factor of two. While this reduction may be efficient, explaining the reason for this choice and the impact on search performance would be beneficial.

**Overall Review:**

Positive aspects:
- Novel approach: The paper introduces an innovative idea in the field of Neural Architecture Search (NAS), focusing on searching for neural architectural building blocks using elementary mathematical operations.
- Detailed methodology: The authors provide a comprehensive description of their approach, including the process of designing a building block, the utilization of the Proximal Policy Optimization (PPO) algorithm, the search for building blocks using a graph representation, and in the appendix, they have reported the values of all the hyperparameters.

Negative aspects of the paper:

- Reason for using PPO algorithm: The authors do not discuss the advantages of the PPO algorithm over other potential reinforcement learning algorithms.
- Impact of increasing the number of total nodes 𝑚: The authors mention that it's possible to increase 𝑚 to adapt to the search progress dynamically, but they do not provide discussion on the potential implications of increasing 𝑚.
- Incomplete results section: The results section seems to be incomplete, as it only reports outcomes related to one of the four datasets mentioned, and it only compares the proposed method to LSTM. Including results for all datasets and a more comprehensive set of building blocks would provide a better evaluation of the proposed method.
- Lack of validation set results: The results in Table 1 only include test set outcomes, and the validation set results are missing. Including validation set results would offer a more comprehensive understanding of the method's performance.
- Insufficient detail on splitting G into sub-graphs: The authors mention that splitting G into sub-graphs and predicting the edges accordingly improves the search for high-performing building blocks. However, they do not provide enough detail on the process of splitting G into sub-graphs or the criteria used for doing so.

**Potential Impact On The Field Of Automl:**

The paper's potential impact on the field of AutoML seems significant. The proposed idea is interesting, and it could be influential in the development of new search strategies and the expansion of the search space in the NAS domain. The method contrasts the current research direction of NAS, which primarily concentrates on computer vision tasks, thereby broadening the scope of the field. However, there are some major concerns in the result section of the paper.

**Review Confidence:**

3: You are fairly confident in your assessment. It is possible that you did not understand some parts of the submission or that you are unfamiliar with some pieces of related work.

**Review Rating:**

7: Weak Accept: Technically sound paper with moderate-to-high impact and strong evaluation, with perhaps some minor flaws.

**Review Summary:**

The paper presents an interesting idea in the field of NAS, proposing an innovative approach for searching neural architectural building blocks using elementary mathematical operations. The proposed idea has the potential to greatly affect how neural networks are designed and optimized. While the idea is intriguing, there are some important concerns that need to be addressed to achieve a more thorough understanding and assessment of the proposed method. The reported results seem to be incomplete, as Table 1 only compares the approaches to LSTM, and among the four datasets, results are only reported for one. Additionally, considering that in Table 2 there is only a small improvement for the fifth dataset compared to all other approaches and some partially good results for the fourth dataset, I decided to weakly reject the current state of the paper.

**Technical Quality And Correctness:**

There are some major concerns about the paper, mainly in the results section of the paper:
- The result section seems to be incomplete, since in the datasets section, the authors mention four datasets: Reber grammar, embedded Reber grammar, adding problems, and memory dataset. However, the results presented in Table 1 are only related to the memory dataset. Is there any reason why the results for the other three datasets are not included?
- It would be beneficial if Table 1 also included the results for the validation set. Currently, the table only presents the test set results.
- Also, I believe that Table 1 should be expanded to include results for other building blocks, not just LSTM, for a more comprehensive comparison. Is there a specific reason why the comparison was only made with LSTM for the memory dataset?
- Lastly, the authors mention that splitting G into sub-graphs and predicting the edges accordingly improves the search for high-performing building blocks. However, they do not provide enough detail on the process of splitting G into sub-graphs or the criteria used for doing so.

---

> ### Author Response · Authors · 2023-05-02
> **Author response to Reviewer NxY5**
>
> We appreciate your recognition of the novelty and potential impact of our work in the field of AutoML. We understand your concerns regarding the results section and the clarity of certain aspects of our paper. In response to your feedback, we have made the following revisions:
>
> 1. Results on synthetic datasets: The Reber grammar, embedded Reber grammar, and adding datasets were employed to assess the general performance of the building blocks in the domain of sequential data during the search process. On the contrary, the memory dataset functioned as the validation dataset to further evaluate the resulting building blocks on an unseen dataset. This evaluation process identified the two highest-performing building blocks, ELENA1 and ELENA2, which we present in this paper. We appreciate and followed your suggestion to extend the table with all ELENAs discovered during the four search processes (see Table 3 in the revised paper). ​​Since the primary purpose of evaluating the resulting building blocks on the memory task was to rank them based on their general performance, we decided to present only the test set results and limit our comparisons to the LSTM architecture. The LSTM just serves as a reference to state-of-the-art building blocks, providing context for the performance of the ELENAs. This approach allows us to maintain focus on the main objective without overextending the scope of the comparison on this synthetic dataset. Instead, we have included an additional real-world dataset to our results, which offers more valuable insights into the performance of the discovered building blocks in a broader context.
>
> 2. Graph splitting details: We have added more information on the process of splitting G into sub-graphs in Section F of the appendix. In essence, this approach simply refers to the order in which the edges of a graph are predicted, without modifying the graph itself. Through various ablation studies, we have empirically investigated the graph neural network architecture, the graph embedding, and the reinforcement learning algorithm, including the edge prediction order. Our results suggest that predicting edges by utilizing sub-graphs leads to the discovery of more complex building blocks earlier in the search process.
>
> 3. Search Strategy: Thank you for pointing out the ambiguity, we have extended the discussion on why we have chosen to employ the PPO algorithm for our search strategy over other potential algorithms.
>
> 4. Number of nodes 𝑚: We have provided more insights into the potential implications of the number of nodes 𝑚 in the computation graph. In general, we have opted for 32 nodes, since such a computation graph can potentially represent most state-of-the-art building blocks. Also, it includes 15 intermediate nodes which proves advantageous to be split into 3 subgraphs to determine the order in which edge types are predicted (see appendix Section F for further details). Additionally, since the number of edges increases quadratically with the number of nodes (formula provided in Section 3.1), choosing a total of 32 nodes provides a suitable balance between the complexity of the graph structure and computational expenses.
>
> 5. Other minor issues: Thank you for pointing out some minor issues. We addressed all of them to further improve the readability of the paper.
>
> With these revisions, we believe we have addressed your concerns regarding the results section and the clarity of certain aspects of our paper. We hope that these changes will provide a more comprehensive evaluation of our method and encourage you to reconsider your assessment. Thank you once again for your valuable feedback!

---

> ### Author Response · Authors · 2023-05-05
> **Follow-Up**
>
> Dear reviewer NxY5,
>
> As the discussion period is coming to an end soon, we would appreciate it if you could evaluate our clarifications and the updated paper.
> If our response resolves your questions and concerns, we kindly ask that you consider raising the rating of our work.
>
> Thank you!

---

> > ### Comment · Reviewer_NxY5 · 2023-05-08
> > **Updated review**
> >
> > I accepted the paper since the majority of my concerns have been resolved, and I believe it is a valuable addition to the existing literature.

---

### Review · Reproducibility_Reviewer_fYxV · 2023-04-13

**Completeness Of Code And Dataset Supplement Rating:** 4
**Usability And Ease Of Reproducibility Rating:** 4
**Actions Required To Increase The Reproducibility And Overall Recommendation:** Review checklist for errors

**Completeness Of Code And Dataset Supplement:**

The supplement is complete, including provided code for all computational components of the paper. The instructions are clear and sufficient for replicating the results. While there was not sufficient time to replicate the architecture search, I was able to verify their evaluation code worked as described.

**Overall Reproducibility Review:**

Overall, the work is easily reproducible given the provided supplement, however there were a few minor errors in the submission checklist.

**Review Confidence:**

4: You are confident in your assessment, but not absolutely certain. It is unlikely, but not impossible, that you did not understand some parts of the submission or that you are unfamiliar with some pieces of the code or data.

**Review Rating:**

9: Strong Accept, all aspects of this are easily reproducible.

**Review Summary:**

n/a

**Summary Of Necessary Code And Dataset Supplement:**

The supplement provided includes the code to run the neural architecture search method developed by the authors, ELENAS, along with separate code to evaluate the models discovered (whose architectures are also included in the supplement) as well as baseline and competing models on two different data sets. The data required, a synthetic memory data set and Tox21, are also included in the supplement.

**Usability And Ease Of Reproducibility:**

Yes, I was able to run the code and reproduce the results. Could improve accessibility by providing both single and multi-gpu configurations for the search pre-set for the user.

---

### Official Review · Reviewer_y13y · 2023-04-13

**Potential Impact On The Field Of Automl Rating:** 3
**Technical Quality And Correctness Rating:** 3
**Clarity Rating:** 3
**Actions Required To Increase Overall Recommendation:** More experiments -- see discussion ab…

**Summary Of Contributions:**

The authors introduce ELENAS, a search method for constructing neural network building blocks based on RL + GNN search over computation graphs, consisting of nodes representing learnable parameters and edges representing elementary mathematical operations. Constructed computation graphs are then mapped back to architectures for performance estimation on synthetic datasets, serving as the reward function. Empirically, they show that ELENAS can design novel building blocks with strong predictive performance, in this case in the domain of sequential data.

**Clarity:**

The paper is easy to read and clear. Some specific comments:

- "Also, we empirically observed that adding the information position of a node in $G$ – with the use of trigonometric functions – as well as of an edge in $G$ – namely whether an edge has already been predicted – substantially improves the overall search process." --> What does it look like to use a trigonometric function to encode information position?
- It would be helpful to include a figure for Section 3.1 to understand the purpose of each type of node

**Overall Review:**

Strengths:
- Intuitive search process that is grounded in strong algorithmic foundations (PPO and GNN)
- Searched architectures, ELENA1 and ELENA2, demonstrate empirically strong performance on both synthetic and real-world datasets -- while being much smaller (in terms of # of parameters) than contemporary methods

Weaknesses:
- Limited experiment scope
    - Only graphs with 32 nodes
    - Only consider sequential data domain
    - Not obvious how to extend synthetic dataset generation to e.g. image classification
- In results, no comparison with Transformer-based models


**Potential Impact On The Field Of Automl:**

The core idea is certainly interesting and novel, and is worth being introduced to the AutoML community -- the question is how easily will ELENAS generalize to other learning tasks, and if novel building blocks emerge there as well.

**Review Confidence:**

4: You are confident in your assessment, but not absolutely certain. It is unlikely, but not impossible, that you did not understand some parts of the submission or that you are unfamiliar with some pieces of related work.

**Review Rating:**

7: Weak Accept: Technically sound paper with moderate-to-high impact and strong evaluation, with perhaps some minor flaws.

**Review Summary:**

The authors introduce ELENAS, a straightforward and reasonable search method to develop novel NN architecture blocks. They demonstrate strong results in the domain of sequential data. While experimental results hint at their method's usefulness, more comprehensive studies are lacking. Overall, an interesting idea with high impact potential, if more sufficiently studied.

**Technical Quality And Correctness:**

The proposed search method is sound and grounded in reasonable prior art for search/graphs. It is a good empirical paper -- the key experiment is interesting and hints at their method's potential, and the authors don't overclaim in their analysis. However, broader experimental work is of course desired.

---

> ### Author Response · Authors · 2023-05-02
> **Author response to Reviewer y13y**
>
> Thank you for recognizing the novelty and potential impact of our work. We appreciate your feedback and agree that broader experimental work is desired. In response to your concerns, we have expanded our experimental evaluation and addressed the specific comments:
>
> 1. Experiments: We additionally tested ELENA1 and ELENA2 on the real-world Penn Treebank dataset. The results further demonstrate the adaptability and effectiveness of our method.
>
> 2. Information on Position Encoding: We have clarified the use of trigonometric functions to encode information position in Section F of the appendix. In essence, the position encoding is inspired by the Transformer model's positional encoding, which uses trigonometric functions (sine and cosine functions) to encode the position of tokens. In this work, positional encoding is just a single additional node feature that is constructed using the sine function.
>
> We hope that the additional experiments and improvements in the manuscript address your concerns and demonstrate the effectiveness and generalizability of our proposed method. As hinted at in the Limitations section, we will conduct further experiments in subsequent work, exploring different domains and search spaces. Regarding image classification, synthetic datasets could for instance be small images with simple geometric shapes such as circles, squares, or triangles optionally with varying colors, sizes, positions, or noise.
>
> Thank you once again for your valuable feedback!

---

### Official Review · Reviewer_v8Xm · 2023-04-14

**Potential Impact On The Field Of Automl Rating:** 3
**Technical Quality And Correctness Rating:** 3
**Clarity Rating:** 3

**Summary Of Contributions:**

This paper proposes Elementary Neural Architecture Search (ELENAS), a method that learns to combine elementary mathematical operations to form new building blocks for deep neural networks. These building blocks are represented as computational graphs, which are processed by graph neural networks as part of a reinforcement learning system. In a set of experiments, the authors demonstrate that their method leads to efficient building blocks that achieve strong generalization and transfer well to real-world data. When stacked together, they approach and even outperform state-of-the-art neural networks at several prediction tasks. Using the proposed method, authors present the building blocks ELENA1 and ELENA2 for the domain of sequential data.

**Actions Required To Increase Overall Recommendation:**

I would suggest author conduct more empirical experiments, e.g. more datasets, compare with more blocks, etc.

**Clarity:**

Overall the motivation of this paper is clear and the presentation is very clear.

**Overall Review:**

Overall the method proposed in this paper is sound to the reviewer.

strength:
- the paper is well-written and the details are discussed properly
- the motivation  and the overall presentation is clear

limitation:
- this paper lacks sufficient empirical results to support the proposed method, which is the most concerning part to the reviewer. Without extensive experiments, the audience does not have sufficient confidence in applying the proposed method. The reviewer would suggest authors conduct more experiments.


**Potential Impact On The Field Of Automl:**

The found build blocks ELENA1 and ELENA2 obtained slightly worse performance than GRU with half parameters. The parameter-effciency of these two blocks may have a medium impact to AutoML as currently the popular LLM are large.

**Review Confidence:**

3: You are fairly confident in your assessment. It is possible that you did not understand some parts of the submission or that you are unfamiliar with some pieces of related work.

**Review Rating:**

5: Borderline Leaning Reject: Technically sound paper where reasons to reject nonetheless outweigh reasons to accept. Please use sparingly.

**Review Summary:**

Overall the method proposed in this paper is sound to the reviewer. However, the experiment evaluation is very limited, which is not convincing enough to tell whether the proposed method, the found building blocks are generally better than others. Given this limited evaluation, the reviewer leans to not support acceptance of this paper.

**Technical Quality And Correctness:**

The proposed building block construction, the search space and the search strategy are technically sound to the reviewer. The found ELENA1 and ELENA2 blocks are applied to several synthetic datasets and one real-world dataset. Given the method is only applied to one real-world dataset, it is not convincing enough to justify the effectiveness of the found building block and the proposed method.

---

> ### Author Response · Authors · 2023-05-02
> **Author response to Reviewer v8Xm**
>
> Thank you for acknowledging the soundness of our proposed method as well as the clear presentation of our method. We appreciate your feedback and understand your concerns regarding the limited experimental evaluation. Thus, we have taken your suggestions into account and have expanded our experiments by another real-world dataset suggested by one of the other reviewers.
>
> We have additionally tested ELENA1 and ELENA2 on the Penn Treebank dataset to tackle your main concern. Although neural architectures found by other NAS approaches, which we compare against, have been specifically searched for this dataset and our computational resources have been a constraining factor, the real-world experiments further demonstrate the generalizability and parameter-efficiency of our building blocks. While it is true that LLMs currently dominate this field, the need for parameter-efficient models remains significant, especially for real-world applications with resource constraints or latency requirements. We believe that fostering research in parameter-efficient building blocks can also contribute to a more sustainable AI landscape. We hope that this perspective on this matter further emphasizes the potential relevance of our work.
>
> With these additional empirical experiments and analyses, we hope to have addressed your concerns. As hinted at in the Limitations section, we will conduct further experiments in subsequent work, exploring different domains and search spaces.
> Thus, we would highly appreciate it if you considered raising the rating of our work.
>
> Thank you once again for your valuable feedback!

---

> ### Author Response · Authors · 2023-05-05
> **Follow-Up**
>
> Dear reviewer v8Xm,
>
> As the discussion period is coming to an end soon, we would appreciate it if you could evaluate our clarifications and the updated paper. If our response resolves your questions and concerns, we kindly ask that you consider raising the rating of our work.
>
> Thank you!

---

### Official Review · Reviewer_LpEh · 2023-04-15

**Potential Impact On The Field Of Automl Rating:** 3
**Technical Quality And Correctness Rating:** 3
**Clarity Rating:** 4

**Summary Of Contributions:**

This paper presents an RL-based Neural Architecture Search method that uses elementary operations to construct novel building blocks for sequence datasets.  The paper focuses on searching for novel building blocks, rather than reusing widely used NN operations, such as the normalization layer. During the search step, the authors used synthetic datasets of increasing complexity to expedite the search procedure. This is a remarkable contribution.

**Actions Required To Increase Overall Recommendation:**

As also mentioned in Section 5, I highly recommend authors consider evaluating the non-sequential datasets, such as CIFAR-10. This can significantly improve the paper's quality and impact. Please also consider addressing my concerns mentioned in the previous sections.

**Clarity:**

•	The paper is overall well-written and easy to follow. I have some minor concerns that addressing them might improve the presentation of the paper.

o	Line 31: Why are the building blocks used in most NAS approaches generally challenging to design?

o	Line 224 – Line 226: It is hard to follow! I do not understand why other search methods are not the first choice in this paper.

o	The ablation study is missing. For example, it is beneficial to see the impact of the proposed method with and without graph splitting.


**Overall Review:**

Strengths:

o	The manuscript is well-written and easy to follow.

o	Proposing a NAS method that searches for elementary operations (e.g., add, multiply, etc.) to discover novel building blocks.

o	The message of the paper is clear and well-supported.

Weaknesses:

o	Experiments of the paper are very limited to one sequential dataset.

o	The novelty of the paper is marginal, similar problems have been previously addressed in [2].

o	Some parts of the paper need more clarification, such as the procedure of graph splitting.

[2] Real, Esteban, et al. "Automl-zero: Evolving machine learning algorithms from scratch." International Conference on Machine Learning. PMLR, 2020.


**Potential Impact On The Field Of Automl:**

Almost all AutoML routines use similar search spaces and/or operations. On the other hand, this paper proposes to search for very novel building blocks with the hope of finding an extraordinary operation (I feel it is similar to the story of finding the original LSTM, right?).

**Review Confidence:**

5: You are absolutely certain about your assessment. You are very familiar with the related work and checked all the details carefully.

**Review Rating:**

4: Weak Reject: For instance, a paper with minor technical flaws, limited impact, and/or weak evaluation.

**Review Summary:**

This paper emphasizes finding novel building blocks. However, the current version lacks diverse experimental results with deep analysis. Thus, I give this paper a weak rejection, but I will change my evaluation if the mentioned concerns are addressed.



**Technical Quality And Correctness:**

•	Overall, the approach and experimentation are technically correct and sound. However, I have serious concerns about the experiments section. I recommend authors consider other real datasets, such as SST [1].

•	The authors used three synthetic datasets during the search procedure with different complexities. Can these three datasets be used to solve any complex real-world problem? If we want to search for other real-world complex tasks, do we need to update/add synthetic datasets? If we select other datasets, I'm not sure we'll see similar improvement.

•	Computation cost (GPU hours) and carbon footprint of the proposed search method have not been reported.

•	The performance of ELENA2 is extremely more than ELENA1 on synthetic data but in real data, ELENA1 works better. What is the reason for these differences in synthetic and real data for the building blocks?  Which features of ELENA2 improve the performance of synthetic data? And why it doesn’t work well on real data?

[1] Socher, Richard, et al. "Recursive deep models for semantic compositionality over a sentiment treebank." Proceedings of the 2013 conference on empirical methods in natural language processing. 2013.

---

> ### Author Response · Authors · 2023-05-02
> **Author response to Reviewer LpEh**
>
> We thank the reviewer for the detailed feedback and for pointing out the significance of our contribution. In the following, we would like to address your comments and concerns:
>
> 1. Synthetic Datasets: The three synthetic datasets aim to evaluate the general performance of the building blocks, such as remembering a specific input over long time lags in the domain of sequential data. We assume to see comparable performances during a search process if we select other synthetic datasets that assess similar properties. This implies that we do not need to update/add synthetic datasets for other real-world complex tasks. This approach aims to find building blocks that generalize well to a wide variety of real-world complex tasks.
> In lines 56 to 59 and the appendix, we have included a more detailed explanation to clarify the motivation behind using these datasets and how they facilitate the search procedure.
>
> 2. Further building block evaluation: Although the emphasis of this work is on the novelty of the methodology rather than on the resulting building blocks, we acknowledge the importance of evaluating them on a broader range of real datasets. As per your suggestion, we have evaluated the performance of our building blocks on the Penn Treebank dataset and further demonstrated that ELENA1 and ELENA2 are parameter-efficient and generalize well to various real-world data.
>
> 3. Computation Cost: Each search process requires about 100 GPU hours on a single NVIDIA A100 GPU. This amounts to approximately 10 kg of CO2 emissions. Since we conducted four separate search processes, we reported a total of 400 GPU hours in our OpenReview submission.
>
> 4. Building Block evaluation: We appreciate your observation regarding the difference in performance between ELENA1 and ELENA2 on synthetic and real data. The reason for this difference might be rooted in the fact that – compared to ELENA1 – ELENA2 follows the concept of a cell state which is beneficial for memorizing the input at the initial time step for a predefined sequence length. Although ELENA2 performs slightly worse than ELENA1 on the real-world dataset, it still is on par with or even outperforms the other architectures with only ⅕ of the size.
>
> 5. Clarity: Thank you for pointing out the remaining unclarities in our work. We have addressed your specific concerns:
> - Line 31: In this regard, we want to emphasize that current NAS approaches frequently incorporate well-established ML concepts, such as pooling layers, normalization layers, or attention mechanisms. Particularly in the domain of sequential data, related work utilizes fixed highway connections, or LSTM cell-inspired variables, which are not trivial to design in the first place.
> - Lines 224-226: We have rephrased this part to provide better insight into our choice of search strategy. In general, since the big search space of elementary mathematical operations only includes a relatively low fraction of high-performing building blocks, employing an RL agent that acquires knowledge from graph structures to consequently refine the search space, appeared to be a natural choice.
> - Line 162-163: We have revised the sentence in question and provided a detailed explanation of the graph-splitting approach in the appendix. In essence, this approach simply refers to the order in which the edges of a graph are predicted, without altering the graph itself. Through various ablation studies, we have empirically investigated the graph neural network architecture, the graph embedding, and the reinforcement learning algorithm, including the edge prediction order. Our findings indicate that predicting edges based on sub-graphs yields high-performing building blocks earlier in the search process.
>
> In summary, we have addressed your concerns by expanding our experiments, providing deeper analysis, and improving the clarity and presentation of our manuscript. As suggested and also highlighted in the section ‘Limitations’, we will perform our method on non-sequential datasets (such as CIFAR-10) in subsequent work. We hope that these revisions will convince you of the merit of our work and encourage you to reconsider your evaluation.
>
> Thank you once again for your valuable feedback!

---

> > ### Comment · Reviewer_LpEh · 2023-05-02
> > **Response**
> >
> > Thanks for addressing my comments. I just increased the 'Clarity' score to 4, and the 'Technical Quality And Correctness Rating' score to 3 (each one is increased by 1 score). Although the idea of the paper is interesting, I still believe more experiments are required to better evaluate ELENAS. I don't know how well the proposed approach will really scale to larger datasets in view of the huge computational overhead (100 GPU hours on a single NVIDIA A100 GPU for each search process).

---

> > > ### Author Response · Authors · 2023-05-02
> > > **Author response**
> > >
> > > Thank you for your response. We consider one of the main novelties of our method to be that scaling to larger datasets is not necessary during the search process, as we solely utilize synthetic datasets to search for parameter-efficient building blocks that can then be scaled to various datasets of any size and complexity.
> > > Although the entire search process takes about 100 GPU hours, ELENA1 was found at 52% and ELENA2 at 41% of the search process. And since our method has already reached its maximum computational complexity, it requires less than 2 GPU days to find high-performing, general-purpose ELENAs. In comparison, Efficient NAS (ENAS) finds recurrent cells in about 0.5 GPU days, DARTS in about 1 GPU day, and NAS in about 10,000 CPU days (Liu et al., 2018). Given these facts, we would highly appreciate it if you considered adapting your overall review rating as well. Thank you!

---

### Author Response · Authors · 2023-05-02
**Reviewer Summary**

We once again thank the reviewers for their extensive feedback, through which we could further improve the clarity and technical quality of our work. This can be summarized as follows:

1. Potential Impact On The Field Of AutoML: We appreciate the reviewers' acknowledgment of the novelty and potential impact of our work. Reviewers y13y and NxY5 explicitly recognized the significance of our contributions and considered our method as "worth being introduced to the AutoML community." Other reviewers conveyed their receptiveness to our method, provided that we conduct more extensive evaluation of our building blocks. Following their suggestions, we have performed additional experiments on the widely-used real-world Penn Treebank dataset. It is noteworthy that the neural architectures found by other NAS approaches, which we compare against, have been specifically searched for this dataset and that our computational resources have been a constraining factor. Nonetheless, these additional results highlight the parameter-efficiency and generalizability of the building blocks and thus the effectiveness of our method.

2. Technical Quality And Correctness: No major technical issues were identified by the reviewers.
In the revised paper, we have addressed the remaining questions, including the choice of search strategy, the use of synthetic datasets, the size of the computation graph, the utilization of positional encoding, and the approach of graph-splitting. Also, we would like to thank reviewer fYxV for the high rating regarding the reproducibility of our method.

3. Clarity: We appreciate that all reviewers found our approach clearly presented and easy to follow. The remaining clarity concerns could be cleared up as part of the comments as well as in the revised paper.

We hope that our revisions resolve your questions and concerns and that the additional experiments provide compelling evidence of the effectiveness of our method. Thank you for your time and efforts!